# Thirty Years of sRNA-Mediated Regulation in *Staphylococcus aureus:* From Initial Discoveries to In Vivo Biological Implications

**DOI:** 10.3390/ijms23137346

**Published:** 2022-07-01

**Authors:** Guillaume Menard, Chloé Silard, Marie Suriray, Astrid Rouillon, Yoann Augagneur

**Affiliations:** 1CHU Rennes, INSERM, BRM (Bacterial Regulatory RNAs and Medicine), SB2H (Service de Bactériologie Hygiène-Hospitalière), University Rennes, UMR_S 1230, F-35000 Rennes, France; guillaume.menard@chu-rennes.fr (G.M.); marie.suriray@chu-rennes.fr (M.S.); 2INSERM, BRM (Bacterial Regulatory RNAs and Medicine), University Rennes, UMR_S 1230, F-35000 Rennes, France; chloe.silard@univ-rennes1.fr (C.S.); astrid.rouillon@univ-rennes1.fr (A.R.)

**Keywords:** *Staphylococcus aureus*, sRNAs, regulation, targetome, in vivo expression, virulence, metabolism

## Abstract

*Staphylococcus aureus* is a widespread livestock and human pathogen that colonizes diverse microenvironments within its host. Its adaptation to the environmental conditions encountered within humans relies on coordinated gene expression. This requires a sophisticated regulatory network, among which regulatory RNAs (usually called sRNAs) have emerged as key players over the last 30 years. In *S. aureus*, sRNAs regulate target genes at the post-transcriptional level through base–pair interactions. The functional characterization of a subset revealed that they participate in all biological processes, including virulence, metabolic adaptation, and antibiotic resistance. In this review, we report 30 years of *S. aureus* sRNA studies, from their discovery to the in-depth characterizations of some of them. We also discuss their actual in vivo contribution, which is still lagging behind, and their place within the complex regulatory network. These shall be key aspects to consider in order to clearly uncover their in vivo biological functions.

## 1. Introduction

Over the last few decades, bacterial regulatory RNAs have emerged as key players in gene-expression reprogramming. They are embedded in complex regulatory networks to fine-tune the bacterial adaptation to environmental cues. Most often, they are noncoding, of a short length, and defined as highly structured and stable compared with mRNAs [1,2,3]. They are usually defined under two classes based on their genetic localization: antisense-encoded sRNAs or *trans*-encoded sRNAs, which are expressed from intergenic regions (IGRs). However, recent discoveries have shown exquisite modes of biogenesis with the identification of 5′ and 3′UTR-derived sRNAs, which indicates that sRNAs that do not perfectly fit the usual features may no longer be considered exceptions [4]. Therefore, although they are often referred to as sRNAs (for small RNAs), a more appropriate term would be regulatory RNAs, given that they interact with many actors to perform their regulatory functions [5,6]. For simplicity, we will keep referring to them as sRNAs hereafter. To regulate their targets, they act as repressors or activators through the modification of ribosome accessibility and/or mRNA stability [2,6]. The binding of sRNA repressors results in translation inhibition by preventing ribosome loading, or else recruits an RNase for mRNA degradation. sRNA activators, in turn, bind the anti-Shine–Dalgarno sequence to release occluded RBS or protect mRNA-cleavable sites. To date, they are found in nearly all bacteria, such as *Staphylococcus aureus*, and are involved in all biological processes. The major pathogen *S. aureus* is responsible for a large set of infections, ranging from minor skin infections to life-threatening conditions, and some strains are resistant to antibiotics [7]. Its adaptation to the ever-changing environmental conditions requires the coordinated expression of various factors, with some of them being under the control of sRNAs. In *S. aureus*, sRNAs are known to regulate diverse functions, such as metabolism, virulence, and antibiotic resistance [8]. Therefore, they are considered as attractive targets for the development of antistaphylococcal therapies. In this review, we report thirty years of research on sRNAs and our current knowledge. We will first describe the initial discoveries and subsequent rush to identify a multitude of sRNA candidates, as well as the experimental and computational approaches successfully conducted to define the sRNA molecular targets. Then, we will discuss their various functions in *S. aureus*, with a specific focus on their expressions in vivo, and we summarize their interconnections with other regulators to illustrate their integration in complex regulatory networks. Altogether, this overview of thirty years of sRNA studies reveals that a thorough characterization is more complicated than initially anticipated, and that the study of their role in vivo, along with their actual place in regulatory networks, shall be the next challenges to face for an in-depth understanding of their contribution.

## 2. From RNAI to Hundreds of sRNA Candidates

The *S. aureus* sRNA chapter opened in 1989, when RNAI was characterized as an antisense RNA that negatively regulates staphylococcal plasmid pT181 replication [9]. In 1993, the same team discovered RNAIII, the *agr*-system effector, which remains, to date, the most studied sRNA with its major regulatory functions deciphered [6,10,11]. Since then, many sRNAs have been identified using various approaches, and the advent of high-throughput sequencing technologies has led to the identification of hundreds of sRNAs [3,12,13,14]. They are expressed from the core genome or the accessory genome, and they are located within IGRs or are derived from 5′/3′-UTR. Interestingly, the majority of these sRNAs have been identified in *S. aureus* strains, but not in coagulase-negative staphylococci (CoNS), which constitutes an *S. aureus*-specific trait and explains, in part, the higher virulence than CoNS [14]. The chronology and relevant characteristics of the sRNA discoveries are summarized in Figure 1 and Table 1.

Overall, there are two distinct periods in sRNA identification. First, sRNAs were mainly identified using in silico predictions and/or microarrays, with or without further functional characterization [15,16,17,18,19,20].

**Table 1 ijms-23-07346-t001:** Classification and cartography of major *Staphylococcus aureus* sRNA discoveries.

sRNA Family/Name	Signification	Number of sRNA Candidates	Localization	Experimental Approach	Further Validation (Validated)	Strain	Clonal Complex	Reference
RNAI		1	Accessory genome (plasmid)	In silico prediction	Northern blot	NCTC8325	CC8	[9]
RNAIII		1	Core genome		NB, mutational analyses	NCTC8325	CC8	[10,21]
Spr	Small pathogenicity island RNA	7	Accessory genome (pathogenicity islands)	In silico prediction	Northern blot (7)	N315	CC5	[15]
SSR	Small stable RNAs	126	Undefined	DNA arrays (GeneChips)	None	UAMS-1	CC30	[16]
Wan	Wan	8	Undefined	DNA arrays (GeneChips)	Northern blot (8)	N315	CC5	[17]
Rsa	RNA of *S. aureus*	11	Undefined	In silico prediction	Microrray, Northern blot, RACE (11)	Various strains		[18]
RsaO	RNA of *S. aureus* Orsay	48	Undefined	In silico prediction	Northern blot (7)	N315	CC5	[19]
RsaO	RNA of *S. aureus* Orsay	30	Undefined	Pyrosequencing	Northern blot (15)	N315	CC5	[22]
SAU	*S. aureus* ncRNA	142	Undefined	Cloning and sequencing of short cDNAs	Northern blot (18)	A3878 I A3878 III	CC5	[23]
Teg	Transcript from experimental method from Geneva	163	Core genome (154 sRNAs) Plasmid (9 sRNAs)	RNA-Seq	RT-qPCR (26)	N315	CC5	[12]
Sbr	SigB-dependent small RNA	3	Core genome	In silico prediction	Northern blot (3)	Various strains	CC8	[20]
JKD sRNA	«JKD6008» *S. aureus* strains	409	Core genome (360 sRNAs) Accessory genome (49 sRNAs)	RNA-Seq	None	JKD6008 JKD6009	CC8	[13]
Tsr	Tampa small RNA	39	Core genome	RNA-Seq	Northern blot (5)	USA300	CC8	[24]
S	S	48	Core genome	Tiling-array	Northern blot (7)	HG001	CC8	[25]
Srn	Staphylococcal regulatory RNAs	21	Core genome (6) Accessory genome (15)	RNA-Seq	Northern blot RT-qPCR (17)	Newman	CC8	[26]

Then, through tilling array and RNA sequencing (RNA-Seq), a significant number of putative sRNAs were described [12,13,14,22,23,24,25,26]. This ongoing interest in uncovering staphylococcal sRNAs, and the absence of a consensual annotation for sRNAs, resulted in a drastic increase in the number of sequences identified, as well as acronyms. To cope with these issues, the BSRD database, which compiles bacterial sRNA sequences and their annotations, was created in 2013, which was soon followed by the specialized *Staphylococcus* regulatory RNA database (SRD) (http://srd.genouest.org/, accessed on 27 June 2022) [14,27]. From a list of 894 theoretical sRNA sequences, and after meticulous refining, the SRD authors described 575 sRNA sequences devoid of any redundancies [14]. In 2016, a transcriptome analysis in a USA300 strain enabled the annotation of 303 sRNAs, whose expressions were subsequently analyzed [3,24]. Although the number of sRNAs is steadily increasing, the community agrees that it is difficult to assess whether the identified sequences actually correspond to sRNAs, and especially when no validation is performed downstream of RNA-Seq. In 2018, a more specific definition was introduced according to explicit criteria: (i) sRNAs must have their own promoter and transcription terminators, (ii) their transcripts should not overlap with another antisense transcript, and (iii) they are expressed from IGRs (i.e., considered as *trans*-acting sRNAs) [28]. This enabled a reduction in the number of sRNAs to around 50 bona fide sRNAs, among which many are connected to virulence and metabolism regulation.

Most of the sRNAs discovered so far were identified under specific conditions. For instance, some Rsa sRNAs are induced during oxidative stress or cold shock, while some Teg sRNAs respond to oxidative or pH stresses [12,18]. JKD sRNAs display specific expression profiles in response to different classes of antibiotics, whereas few “S” sRNAs were upregulated under media-mimicking host conditions [13,25]. In a recent study, Bastock and colleagues showed a temperature-dependent specific transcriptome profile that included sRNAs [29], which indicates that sRNAs respond to precise triggers. Overall, the burst in sRNA identification was mainly correlated with descriptive studies regarding sRNA expression rather than sRNA function. This was likely due to the fact that specific approaches to identifying sRNA molecular targets were, for a short period, lagging behind. In the next section, we will describe the different types of approaches developed to improve our knowledge of the sRNA functions and mechanisms of action in *S. aureus*.

## 3. On the Quest to Identify sRNA Molecular Targets

The identification of sRNA targets is needed to decipher the regulatory networks and their impact on the adaptation of the bacterium to environmental cues. The search for sRNA targets can be conducted even though an sRNA has not yet been associated with a specific phenotype. Over time, several approaches were developed, from tedious strategies that enabled the monitoring of modifications to the protein content, to more recent high-throughput techniques (Figure 2). These latter allow for the rapid identification of candidates, but they require a thorough sorting of targets that appear as the most relevant prior to experimental confirmation. In parallel, several algorithms were developed to predict and study interactions in silico (Figure 2). These approaches, and the sRNA–mRNA pairings identified, are discussed in the subsections below.

### 3.1. Experimental Approaches

One of the first methods used to identify the primary targets of an sRNA was the 2D DIGE [30]. It involves the separation of proteins from different bacterial compartments based on their charges and molecular weights, and a comparison between wild-type and knock-out strains. For instance, in 2013, Kaito and colleagues showed that the proteins HutU, Spa, and Ddh were increased in a cell lysate of a USA300 strain containing the Psm-mec sRNA [31]. Then, they demonstrated that this increase was due to the pairing of Psm-mec with its direct target, the *agrA* mRNA, whose stability was decreased. Therefore, HutU, Spa, and Ddh are indirect targets of Psm-mec sRNA because they are under the control of AgrA. This was a successful approach to identify a protein target of SprX sRNA [32]. In their study, the authors showed a decrease in SpoVG, which is involved in glycopeptide resistance, in a strain expressing SprX, compared with its isogenic deleted strain.

Although less resolutive, 1D SDS-PAGE can provide substantial information. Protein extracts from different bacterial compartments (extracellular, membrane, cytoplasmic, etc.) can be separated and bands of interest analyzed by mass spectrometry (MS) or N-terminal sequencing. In 2010, the Sbi protein was identified as a target of SprD sRNA after an analysis of the secreted-protein profiles [33]. Indeed, an additional band corresponding to a 45 kDa protein was detected in supernatants taken from a strain lacking the *sprD* gene. A few years later, the same approach was conducted to identify variations in proteins secreted in the *S. aureus* Newman strain. After the SDS-PAGE analysis, the major staphylococcal autolysin (Atl) was found to be repressed when SprC sRNA was expressed [34]. In these examples, the proteins of which the expressions are modulated are not the direct target of the sRNA because regulation actually occurs through the sRNA–mRNA interaction.

Overall, an advantage of these proteomic approaches is that they help with finding the target in a particular condition. However, one main limitation is that they do not allow for an extensive decryption of the targetome because moderate effects may be missed due to a lack of sensitivity, and especially when using 1D separation. Moreover, proteins of which the expressions are modified are not necessarily the primary targets, as revealed in the Psm-mec study.

In recent years, novel techniques based on high-throughput screenings have been developed, which has allowed for large-scale analyses of RNA–RNA or RNA–protein interactions (i.e., their direct targets). These deep analyses devoted to the targetome and pairing-site identification encompass MS2-affinity purification coupled with RNA sequencing (MAPS), Hybrid-trap-seq, Hi-GRIL-seq, RIL-seq, and cross-linking, ligation and sequencing hybrids (CLASH) [35]. Among them, MAPS was successfully used in several *S. aureus* studies. MAPS consists of the fusion of an MS2 tag to an sRNA of interest expressed from a plasmid, followed by total RNA extraction (Figure 3) [36].

Total RNAs are added to a column preloaded with maltose-binding protein fused to the MS2p protein. Through the MS2/Ms2p interaction, the sRNA of interest is retained, along with the RNAs and proteins that are bound to it. These potential targets are eluted and identified by RNA sequencing for sRNA–mRNA interactions, or by MS for sRNA–protein interactions. The main advantage of MAPS is that it does not require the presence of the Hfq protein, of which its role in *S. aureus* has not been demonstrated [37]. This technique was first used in *S. aureus* to study the RsaA targetome [38]. The authors identified mRNAs encoding transcription factor MgrA, the SsaA protein involved in peptidoglycan synthesis, and the anti-inflammatory protein FLIPr, as the primary targets. A similar approach was used to identify the targetomes of RsaI, RsaC, SprY, and RsaG [39,40,41,42]. The use of RsaI as a bait allowed for the identification of mRNAs encoding a permease for glucose uptake (GlcU_2), along with Fn3K and IcaR, which are involved in protection against high concentrations of glucose and the repression of exopolysaccharide production, respectively [39]. RsaC was found to repress the expression of the superoxide dismutase SodA, which is involved in the oxidative-stress response [40]. A recent study of the RsaG targetome revealed mRNAs of the transcription factors Rex, CcpA, and SarA as molecular targets [42]. Interestingly, studies conducted on both RsaI and RsaG showed that these two sRNAs interact together, which suggests complex sRNA regulatory networks (see Section 6 of this review). This was further confirmed with the study on the novel sRNA SprY, which interacts with RNAIII and acts as an RNAIII sponge [41]. To date, this technique appears robust, even though the choice of the culture condition and time of RNA extraction may be determining factors to identify the relevant primary targets of sRNAs.

Hybrid-trap-seq was developed a few years ago and has enabled the identification of RsaE targets [43]. The principle is to perform an in vitro transcription of an sRNA of interest, which is biotinylated at the 3′ end, and immobilized on magnetic streptavidin beads (Figure 2). Synthetic sRNA is mixed with a pool of denatured total RNAs, which can be extracted from different culture conditions. After the hybridization and washing steps, RNAs are eluted and sequenced to identify RNA targets. Using RsaE as a bait, Hybrid-trap-seq revealed that the *rocF*-mRNA-encoding RocF, which is involved in arginine catabolism, is a direct target [43].

Other techniques based on high-throughput sequencing were developed to study bacterial sRNAs in various species. For instance, RIL-seq (RNA interaction by ligation and sequencing) enables the identification of RNA targets through the formation of in vitro RNA–RNA complexes, which are purified and sequenced. Until now, this technique was only used in Gram-negative bacteria because it requires Hfq to recover the RNA–RNA complexes [44]. In *S. aureus*, this RNA matchmaker is not needed for sRNA-mediated regulation, which makes this approach unsuitable. GRIL-seq [45], and its Hi-GRIL-seq [46] variant, which are not dependent on Hfq, exist but have never been used in *S. aureus*. Similarly, CLASH enables the identification of the RNA–RNA complexes bound to the RNase E in *Escherichia coli* [47], but it has not yet been used in *S. aureus*.

### 3.2. Computational Tools

Several in silico approaches that allow for the computational search for targets of an sRNA have been reported and reviewed in a book chapter [48].

IntaRNA (http://rna.informatik.uni-freiburg.de/IntaRNA/Input.jsp, accessed on 27 June 2022) allows for testing the strength of interactions between two RNAs based on three different criteria: (i) the hybridization energy between the two molecules, (ii) the energy necessary to unfold the sRNA, and (iii) the energy necessary to unfold the target allowing for the accessibility of the two regions. An energy balance is then calculated, and it allows us to rank the most likely interactions from a thermodynamic point of view [49]. IntaRNA can be used in two ways: either as a general target finder based on the genome of the chosen strain and the sRNA bait sequence, or to specifically study an interaction with an already anticipated target. It was used to find regions of interaction between Teg41 and *psmα* mRNA [50], and between RsaD and *alsS* mRNA, which allowed for hypotheses regarding the mechanism of action [51]. Then, the targets identified needed further experimental confirmation, such as the gel retardation assay (EMSA), Northern blot, Toeprint, or double-plasmid assays.

RNApredator (http://rna.tbi.univie.ac.at/RNApredator, accessed on 27 June 2022) is used for the same purpose. It allows for the selection of a very large number of genomes and plasmids from different species as a basis, and the accessibility of the target is considered during the search. It uses the RNAplex, which is a dynamic programming approach, to calculate the strength of an sRNA–mRNA interaction [52].

Alternatively, CopraRNA (comparative prediction algorithm for small RNA targets) (http://rna.informatik.uni-freiburg.de/CopraRNA/Input.jsp, accessed on 27 June 2022) compares the conservation of RNA–sRNA interactions in different strains of the same bacterial species using the IntaRNA algorithm. If an interaction is often found in different strains, it will be assumed as very likely. The algorithm allows for different things: the identification of targets for sRNAs, the prediction of interaction domains, and the construction of regulatory networks [53]. The advantage of CopraRNA is that it provides a comparative and predictive dimension between strains, which is not the case for IntaRNA.

TargetRNA2 (http://cs.wellesley.edu/~btjaden/TargetRNA2/, accessed on 27 June 2022) is another powerful algorithm to identify sRNA targets. It uses different criteria, such as the secondary structure of the RNA, the secondary structures of different mRNA targets, the conservation in other bacteria, and the hybridization energy between the sRNA and targets. It can also integrate RNA-Seq data to consider codifferential gene-expression variations, and can therefore improve the accuracy of prediction [54].

These programs can be used in cooperation to collate data and refine the target search. This allows for the selection of interesting targets for experimental validation. For instance, three of these packages (IntaRNA, TargetRNA2, and RNApredator) were used to identify the targets of RsaD [51]. This enabled the discovery of *alsS* mRNA, which was subsequently experimentally confirmed, along with the mechanism of regulation. Overall, computational and experimental approaches can be complementary, as they both have strengths and weaknesses, which were recently discussed by Georg and colleagues using RsaA as an example [55]. These approaches paved the ground for extensive investigations to examine sRNA functions.

## 4. Insight into the *Staphylococcus aureus* RNome and Its Functions: Metabolism, Virulence, and Antibiotic Resistance

Among the hundreds of putative *S. aureus* sRNAs identified, around fifteen were characterized for their related targets and biological functions, which span virulence, antibiotic resistance, and metabolism. A majority are defined as bona fide sRNAs (RNAIII, SprD, SprX, SprY, RsaA, RsaC, RsaD, RsaE, RsaI, SSR42) [28]. These *trans*-encoded sRNAs usually regulate more than one mRNA target, which thus allows for their implication in large sets of bacterial regulation. Until recently, they were considered as strictly intracellular molecular actors, but recent work on extracellular vesicles suggests that they could be involved in long-distance host–pathogen interactions [56]. Although there is limited evidence for their role in host–pathogen interactions in *S. aureus*, the existing data prove their critical regulatory functions and their integration into the complex regulatory network. In addition to RNAIII, for which a large set of targets and regulatory mechanisms are reported (see below), there is a heterogeneity in the characterization of the different sRNA candidates. Here, we present the main biological functions of these sRNAs, and we sort them according to their initial acronyms. The main data regarding these sRNAs are reported in Table 2 and are described in the following subsections (Table 2).

### 4.1. RNAIII

Initially discovered in 1989 as the transcript encoding δ-hemolysin, RNAIII took the spotlight when Novick discovered its regulatory function [10,21]. To date, RNAIII is the most studied and characterized sRNA, and it is the main effector of the quorum-sensing *agr* system [6,11]. It is a 514 nt-long sRNA composed of 14 stem-loops, some of which are rich in cytosine residues, which thus facilitate interactions with the guanine-rich RBS of target mRNAs. Another feature of RNAIII is its dual function, as it encodes for the PSM-toxin δ-hemolysin [57]. Overall, the *agr* system is responsible for the direct or indirect activation and repression of numerous virulence protein-coding genes. At least 138 genes are regulated by this two-component system (TCS) [58,59], while, so far, RNAIII is reported to directly control the expressions of nine targets. It acts as a post-transcriptional regulator that modifies through base-pairing translation and/or target stability in an RNase III-dependent manner [11]. Its mRNA targets encode virulence factors (Hla, Spa, SA1000, LytM, Coa, Eap, Sbi) or global regulators (Rot, MgrA), whose functions have been described by others [11,60,61,62,63,64,65,66,67]. In addition to regulating a large number of targets, RNAIII is a key node in the complex regulatory network (see Section 6 for further details), with links with other riboregulators, such as SprD, ArtR, RsaA, Psm-mec, and SprX [38,57,66,68,69].

The effect of RNAIII on virulence is commonly associated with the transition from colonization to infection via the activation of the quorum-sensing system. This RNAIII-dependent transition leads to a permutation in the expression of virulence factors: the repression of adhesion factors in favor of toxins and other secreted proteins. RNAIII implication in virulence is no longer in question because it, or more precisely, the *agr* system, contributes to the pathogenesis of *S. aureus*, as demonstrated in multiple investigations [11,70]. This includes virulence in murine skin-infection models, in rabbit models of endocarditis and osteomyelitis, and in an intracranial mouse model [71,72,73,74,75,76]. However, its precise function during infection seems difficult to interpret and is a matter of debate. Although the *agr* system, and therefore its effector, is not active during colonization, it is not necessarily active at all stages of infection. It is expected to have an important role during the acute phase of the infection, but less during chronic infections [11,77]. In fact, this is supported by a lower RNAIII expression level in the small-colony-variant (SCV) phenotype than in the wild-type strain, considering that the SCV phenotype is often encountered in chronic and persistent infections [23,78]. Biofilm infections and/or intracellular persisters tend to adopt another lifestyle, in which adhesion and escape immune factors are primordial, consistent with the low activity of *agr*, and therefore RNAIII as observed during nasal colonization [79,80,81]. *S. aureus* with a dysfunctional *agr* system lead to enlarged biofilms and are more resistant to neutrophil attack [82]. This feature was supported in a rabbit infection model where mortality with the *agr*-deleted strain was higher in catheter-associated infection compared with noncatheter-associated infection [83]. Therefore, to better define these issues, it was proposed to differentiate the *S. aureus* evolution into two distinct patterns: the defensive mode, where *S. aureus* expresses preferentially defensive genes, such as adhesins and escape immune proteins, and the offensive mode, in which the bacterium mainly secretes toxins [84].

### 4.2. Spr sRNAs

Sprs were identified as expressed from pathogenicity islands and were therefore predicted to be involved in virulence [15]. Apart from SprB, for which a function remains to be discovered, all other Sprs have been characterized.

#### 4.2.1. SprD

SprD was the first sRNA of the Spr family to be identified as participating in *S. aureus* virulence [33]. However, only one target is known so far: the Sbi protein, which plays a role in adhesion and immune evasion [85]. SprD binds the 5′ of *sbi* mRNA to form a duplex, which results in translation inhibition through the masking of the RBS. In vivo, it appears to be a major regulator of virulence because, in a mouse infection model, the mortality was attenuated in a ∆*sprD* mutant strain, and a decrease in renal abscess formation was observed [33]. This phenotype cannot be explained only by the action on Sbi, which suggests that other unidentified targets play a critical role in vivo. The same authors demonstrated in vitro that Sbi is coregulated by another sRNA: RNAIII [66]. SprD and RNAIII both repress *sbi* mRNA translation, but their action is growth-phase-dependent: SprD acts at the early stage of the exponential phase, while RNAIII acts in the stationary phase. No direct interaction between SprD and RNAIII has been demonstrated.

#### 4.2.2. SprC

In contrast to SprD, SprC was characterized as an antivirulent sRNA [34]. In an in vivo murine infection model, it decreased the virulence, as its deletion led to increased virulence and bacterial dissemination. In the same study, the authors identified one target: the *atl* mRNA, which is a major autolysin, and they deciphered the mechanism of translation inhibition by preventing ribosome loading. This regulation is probably partially responsible for the decrease observed in the *S. aureus* internalization by the macrophages. It was also shown that SprC expression is under the control of the transcription factor SarA, which binds the *sprC* promoter site to prevent its transcription [86]. Subsequently, other potential SprC targets were proposed using bioinformatics approaches [87]. SprC is associated with 44 proteins involved in many biological processes, which suggests the pleiotropic effect of SprC in staphylococcal regulation. It is noteworthy that no direct interaction between SprC and these targets has been provided. Surprisingly, the dogma that SprC is an antivirulent sRNA was not supported in the *Galleria mellonella* infection model, which showed the provirulent effect of this sRNA [88].

#### 4.2.3. SprX

SprX was initially identified as an sRNA that is implicated in bacterial resistance and, more precisely, in glycopeptide resistance [32]. By direct interaction with the RBS of *spoVG* mRNA, SprX inhibits its translation and decreases glycopeptide resistance. Besides this role, it was shown that the use of different biocide exposures can alter the *sprX* expression [89]. The variations were dependent on the biocide type and growth phase. A second SprX target was further identified: the *ecb* mRNA encoding a SERAM [90]. By binding the RBS of *ecb* mRNA, SprX again inhibits its translation. Although no virulence studies were performed, this mechanism suggests the role of SprX in virulence because Ecb is involved in adhesion and immune escape [91]. This hypothesis was latter verified in a mice model of infection [92]. The authors also showed that SprX upregulates the expressions of two major virulence factors by direct interaction, ClfB and Hld, although the precise mechanism of action was not elucidated. Additional analyses have confirmed the key role of SprX in *S. aureus* pathogenesis [69]. SprX indirectly upregulates several autolysins, such as AtlA and LytM, which are considered major virulence factors. An in silico analysis and EMSA demonstrated that SprX targets *walR* mRNA, a positive regulator of autolysins, which supports a walR activation, but the specific mechanism was not uncovered. SprX also plays a role in in vitro biofilm formation, which validates walR activation because the WalKR TCS promotes biofilm formation and *S. aureus* autolytic activity [69,92,93]. However, it is unclear why SprX was considered as a provirulent sRNA in a systemic mouse infection model because the WalKR TCS reduced the mortality and increased the bacterial clearance [92,93].

#### 4.2.4. Spr sRNAs and Toxin–Antitoxin Systems

Studies that focus on the other Sprs actually reveal that they belong to type I toxin–antitoxin systems (TASs) [94,95,96,97]. TASs are frequently distributed among bacteria, including *S. aureus*, and are recognized as important players in physiological homeostasis [98]. Type I TASs are described in *S. aureus* and consist of a protein toxin and sRNA antitoxin that prevent its expression [98,99]. Interestingly, the sRNA antitoxin SprF1 not only locks SprG1 toxin expression with its 3′ end, but also attenuates bacterial translation during hyperosmolar stress through its ability to bind to ribosomes using its 5′ end [100]. This particular mechanism is responsible for persister cell formation, as evidenced by in vitro experiments. This novel discovery may serve as a starting point for the development of sRNA-based therapies if it is confirmed in vivo (within the host). Potential SprF1 targets were also screened [101]. Few proteins were evidenced as potential targets of SprF1, as their expressions were downregulated when SprF1 was overexpressing. Some are glycolysis-related proteins, of which three have reduced the transcript levels in intracellular persisters [101,102]. In silico analyses predicted a potential interaction between SprF1 and the *ppiB* mRNA target, but this hypothesis was not further validated by the EMSA experiments [101].

#### 4.2.5. Spr sRNAs and the Sponge Mechanism

The direct interaction between two sRNAs, which is called the RNA sponge activity, represents a new feature in sRNA-mediated gene regulation, and little is known about this in *S. aureus* [6]. Recently, a sponge mechanism was described in *S. aureus*-pathogenesis regulation, implying RNAIII and a novel sRNA named SprY (alias S629) [41]. SprY tightly monitors RNAIII activity by the base-pairing mechanism and, consequently, modulates RNAIII target expressions, such as the *rot* and *ecb* mRNA. In vitro, SprY reduced hemolysis and pathogenicity in a mouse model of infection. Through this titration, *S. aureus* pathogenicity is reduced, but the conditions for SprY expression within the host have not yet been clarified.

### 4.3. Rsa Family

The Rsa sRNA group includes several sRNAs that are implicated either in virulence and/or in metabolism [103,104]. To date, it is the sole sRNA group for which extensive data are available regarding the targetome through the implementation of MAPS [36,105]. Consequently, some direct sRNA targets were discovered in vitro for RsaA, RsaI, RsaC, and RsaG [38,39,40,42].

#### 4.3.1. RsaA

RsaA was the first *S. aureus* sRNA characterized as an antivirulent sRNA. In a mouse model of infection, RsaA attenuated acute systemic infection and enhanced chronic catheter infection (e.g., biofilm formation). In vitro, it led to biofilm formation and decreased capsule-protein production, which resulted in a greater sensitivity to neutrophils. It also represses the expression of the master regulator MgrA through translation inhibition and enhances mRNA degradation, which is the opposite action compared with RNAIII. Besides MgrA, two other RsaA targets were identified by MAPS: *ssaA*-like and *flr* mRNAs, which encode proteins that are mostly implicated in biofilm formation and the immune-evasion system [38,106,107]. RsaA is able to form duplexes with these mRNA targets and subsequently inhibit translation initiation [38]. Others adhesins, such as Spa, SasG, Ebh, or ClfB, were also recognized as indirect targets, and appeared to be positively regulated due to MgrA repression. Together, these findings are consistent with the crucial role of RsaA in biofilm formation.

#### 4.3.2. RsaC

RsaC is considered an uncommon sRNA. It results from the maturation of the 3′UTR of the mntABC operon, and its length is strain-dependent [40]. It contributes to the regulation of oxidative stress, mostly through the translation inhibition of *sodA*. Under manganese starvation, RsaC is highly functional and inhibits unnecessary SodA production, which is an Mn-dependent detoxification enzyme. Instead, it can indirectly trigger the synthesis of SodM, which thus promotes a dual effect: the detoxification of reactive oxygen species, and the preservation of the manganese availability. Panthee et al. subsequently demonstrated the in vivo impact of RsaC in *S. aureus* pathogenesis [108]. An RsaC-deleted strain was shown to be less virulent in a mouse systemic-infection model and, interestingly, the bacterial burden was decreased only in the heart. In vivo, RsaC seems to regulate many genes that are involved in either metal acquisition or virulence, and they block neutrophil activities, which is in agreement with the observations of Lalaouna and colleagues [40,108].

#### 4.3.3. RsaE

RsaE is a global sRNA regulator whose expression is SrrAB-, and, to a smaller extent, it is *agr*-dependent. It is implicated in metabolic adaptation [18,43]. RsaE regulates the synthesis of at least 86 proteins both directly and indirectly. It was proposed to interact with target mRNAs through its UCCC motif, and, consequently, to inhibit the formation of the active ribosomal complex [18]. The RsaE targetome was deciphered, thus confirming previous results, but highlighting differences in the targetome according to the methods used (transcriptomic versus Hybrid-trap-seq experiments) [43]. *rocF* mRNA, one of the common targets, was especially studied, and the authors demonstrated that RsaE inhibited translation through the binding of the SD sequence [43]. Finally, in vitro experiments indicated that RsaE is a repressor of amino acid catabolism. There is no direct evidence that supports that RsaE participates in *S. aureus* virulence, but, in *Staphylococcus epidermidis*, RsaE was shown to repress IcaR synthesis by interacting with the 5′-UTR of the *icaR* mRNA [109]. The polysaccharide intercellular adhesin (PIA), a peptidoglycan component, is crucial for adhesion, immune-system escape, and biofilm biogenesis, and its biosynthesis is negatively controlled by the IcaR repressor [110,111]. RsaE actually enhances biofilm formation by both promoting PIA synthesis, and the release of extracellular DNA in *S. epidermidis* [109].

#### 4.3.4. RsaD

RsaD is commonly associated with the regulation of genes that are involved in nitric oxide stress and cell density, based on its expression profile [18,39,112]. However, its precise regulatory function was not clarified until recently. In 2020, RsaD was shown to belong to the CodY regulon [51]. Transcriptomic analysis and EMSA demonstrated the repressor effect of CodY upon RsaD expression. Then, using in silico approaches, one RsaD direct target was further identified, which was, namely, *alsS* mRNA, and the mechanism of the translation inhibition was characterized. These data, along with the previous study showing that CodY regulates *alsS* expression, indicate the contribution of RsaD in carbon metabolism [51,113]. By inhibiting the enzymatic activity of AlsS (α-acetolactate synthase), RsaD leads to the increased production of acetate from pyruvate, which is a metabolite that potentiates cell death. Because pyruvate is a product of glucose, it makes sense to assume that, when glucose is in excess, RsaD must be repressed to reroute the pyruvate metabolism to the production of acetoin instead of acetate. The central metabolite pyruvate was shown to modulate *S. aureus* virulence by promoting leucocidin formation; thus, it may be necessary to clarify whether RsaD can be considered an antivirulent sRNA because it tends to modulate pyruvate conversion [51,114].

#### 4.3.5. RsaI, RsaG, and Nascent Interconnections

RsaI is repressed during glucose overabundance by the catabolite control protein A (CcpA), and is derepressed under glucose starvation [39]. At least three RsaI direct targets were identified by MAPS: *glcU_2*, *fn3K,* and *icaR* mRNAs. They encode a glucose-uptake protein, a detoxification protein, and the repressor of exopolysaccharide production, respectively. In all cases, RsaI represses translation by base-pairing with the SD sequence (*glcU_2*, *fn3K*) or with the 3′ UTR (*icaR*). It is likely that RsaI, by repressing IcaR, induces biofilm formation, although it has not been demonstrated.

RsaG is the last Rsa sRNA characterized so far [42]. RsaG is highly expressed in response to extracellular glucose-6-phosphate (G6P), which is the activated form of glucose [39], but it is not essential for G6P uptake and catabolism [42]. RsaG appears to be critical when *S. aureus* is internalized into host cells or in the presence of mucus-secreting cells, which thus enables the bacterial fitness to different environmental conditions. Several RsaG targets were identified by MAPS, and, among them, transcriptional factors were predominant, such as Rex, Sar family proteins, and CcpA [42]. In their study, the authors showed that RsaG is able to bind the 5′-UTR, SD sequence, or RBS of mRNA targets, leading to the translation initiation, degradation, or stabilization of mRNAs. However, the actual contribution of SarA and Ccpa protein expression remains to be elucidated.

It was proposed that RsaI could be a central node because of its interaction with the other sRNAs, such as RsaD, RsaE, and RsaG [39]. This was partially elucidated because it was shown that RsaI binds RsaG, which secondarily led to a modification of RsaG-dependent interactions [42]. This finding reinforces the assumption of RNA sponge activity and illustrates the *S. aureus* complex regulatory network, including all the hierarchical regulation levels. Although nothing indicates direct links between RsaD, RsaE, RsaG, RsaI, and virulence regulation, there is a common feature between these sRNAs: they are important to the adjustment of metabolic pathways, and, thus, to the adaptation to specific host microenvironments. RsaD, RsaI, and RsaG are intricate, as they are linked with other’s metabolite-responsive global regulators, such as CodY, CcpA, and Rex [39,42,51], which are known to be connected with metabolism and virulence [115]. Therefore, infection studies in animal models could provide new insights regarding their implications, and especially during biofilm biogenesis or internalization, where the metabolic conditions are altered. This idea is supported by past studies, which showed that alterations in the central metabolic pathways caused an RNAIII-dependent regulation, and virulence-factor-expression modifications [116,117,118].

Recently, RsaF was shown to downregulate two mRNA protease targets: *hysA* and *splD*, and to contribute to biofilm biogenesis [111,119]. When RsaF is disrupted, in vitro biofilm production is enhanced, which supports the idea that RsaF negatively controls biofilm formation [119]. The direct interaction between RsaF and mRNA targets was validated by an in silico prediction and EMSA, although the exact mechanism of regulation was not provided.

### 4.4. Teg Family

Following their initial discovery in 2010 [12], only two (Teg49 and Teg41) have been characterized [50,120], and both were found to play a role in *S. aureus* pathogenesis. Initially, they were identified in the 5′ UTR of the global regulator SarA and at the locus encoding for αPSM peptides, respectively. These loci suggested a putative link with the *S. aureus* infection process. Teg49 expression is modulated by the polymerase factor SigB or the transcription factor CshA, and it seems to act on *sarA* transcription through a stem-loop structure called HP1 [120,121]. No effect on staphylococcal virulence was demonstrated in vivo, although a mutation in HP1 leads to a decrease in both SarA protein and *RNAIII* mRNA, with a reduced capacity to form biofilms in vitro. Recently, 22 novel mRNA targets were identified by RNA-Seq as putative Teg49 targets in a screen that discarded previously known SarA targets [122]. Among them, *spn* was repressed by Teg49 at the post-transcriptional level. *spn* encodes for the staphylococcal peroxidase inhibitor (SPIN), which inhibits the myeloperoxidase of neutrophils [123]. A Teg49 mutant was shown to be more resistant to neutrophil effects, and exhibited a prolonged intracellular survival within neutrophils [122]. In addition to the modulation of biofilm formation, Teg49 is therefore important to the modulation of the innate immune response in a SarA-independent manner. By deletion and/or overexpression experiments, Teg41 was shown to positively regulate αPSM production, and thereby *S. aureus* hemolytic activity [50]. In silico analyses predicted the binding of the 3′ end of Teg41 on the fourth gene of the αPSM operon, but no direct in vitro interaction was assessed to confirm this. In vivo, Teg41 implication in virulence was evidenced in a mouse model of abscess infection [50].

### 4.5. Other sRNAs

In 2013, a novel sRNA, ArtR, was described [68]. ArtrR is under the control of the *agr* system, with AgrA acting as a direct repressor of ArtR transcription. The *agr* system effects could therefore be mediated by both RNAIII and ArtR sRNAs. ArtR positively modulates the expression of α-hemolysin by an indirect mechanism, through base-pairing the 5′ UTR of *sarT* mRNA, which results in translation arrest and mRNA degradation. Consequently, the authors speculate that ArtR is involved in virulence regulation, even though there is no direct evidence in infection models. Because ArtR and RNAIII both upregulate α-hemolysin production, and AgrA represses ArtR, it is suggested that α-hemolysin production may be increased in *agr*-deficient strains [124].

SSR42 is also another sRNA that participates in *S. aureus* virulence [125,126,127]. It is predominantly expressed during the stationary growth phase and is under the control of the transcription factor (TF) Rsp [125,126]. Transcriptional mRNA profiles indicated that SSR42 affects the expressions of 82 mRNAs, with the repression of 80 of them [125]. These targets include major virulence determinants, such as Spa and Sbi, which are also RNAIII targets. To support these findings, it was demonstrated that SSR42 was required for in vitro hemolysis activity and enhanced skin abscesses in a murine infection model [125]. The precise mechanism has not been entirely elucidated, but SSR42 seems to act through intermediate molecules that subsequently regulate the expressions of virulence factors. This possibility was further validated, which thereby integrates SSR42 into the complex regulatory network and highlights the intricate interactions with others regulators [127]. It was shown that SSR42 belong to several regulons, such as AgrA, ArlRS, SaeS, and CodY, and that this sRNA modulated *hla* expression in a SaeRS-dependent manner. Even though the entire concept has not been demonstrated yet, SSR42 would favor the expression of toxins to the detriment of surface proteins, as already evidenced for RNAIII.

The *psm-mec* gene is located on the staphylococcal cassette chromosome *mec* (SCC*mec*), which encodes methicillin resistance. The *psm-mec* locus is considered a major virulence determinant that constitutes a key hub in both virulence and antibiotic resistance [128]. It was first demonstrated that the F region from the SCCmec containing the psm-mec ORF was associated with a reduced mortality in a mouse model of systemic infection [129]. The pathophysiological processes observed were lowered PSMα production and a decrease in bacterial dissemination in favor of biofilm formation. Interestingly, the *psm-mec* transcript product was responsible for reduced PSMα production, whereas bacterial dissemination and biofilm formation were dependent on both *psm-mec* RNA and the resulting peptide. *psm-mec* RNA is now considered as a dual sRNA because it codes for a 22 amino acid peptide, the cytolysin PSMmec, and it has a regulatory function by targeting the coding region of *agrA*, and then repressing *agrA* translation [31,57]. Among methicillin-resistant *Staphylococcus aureus* (MRSA) isolates, two entities are described: hospital-acquired MRSA (HA-MRSA) and community-associated MRSA (CA-MRSA), and there is clear evidence that CA-MRSA are more virulent than H-MRSA [7,130]. One reason would be the lack of psm-mec function in CA-MRSA, which thus explains the high-virulence properties compared with HA-MRSA [31,129]. However, these statements must be tempered because virulence regulation by the *psm-mec* locus is strain-dependent [131].

Altogether, this section summarizes the current knowledge on the *S. aureus* sRNA function through mostly in vitro studies and an examination of the deletion or overexpression of an sRNA in an animal model to link it with virulence. However, for a better understanding of their role, or to unravel their unanticipated functions, an in-depth characterization of their expression in vivo is critical.

## 5. In Vivo sRNA Expression in Humans and Animal Models

Until the 2000s, seminal studies to prove the effects of *S. aureus* virulence factors were conducted in vivo using mainly two strategies: gene inactivation, or the adaptive gene transfer expression of a selected factor [132]. Thus, global regulators, including sRNAs such as RNAIII, have been identified as virulent factors in vivo. For example, it was demonstrated that *agr* mutants were less virulent than wild-type strains in infection models of arthritis or osteomyelitis, in rabbits and mice [133,134]. Although this approach was interesting, understanding the in vivo interaction and regulation was not possible [132,135]. It was then proposed to screen the in vivo transcription profile of the *agr* system and RNAIII, which was the only sRNA known at that time [136,137]. The results were surprising for several reasons: (i) RNAIII was poorly expressed in vivo, (ii) there was no correlation between the RNAIII expression and the bacterial burden, and (iii) there were discordances in the RNAIII target expression. This was the first description of the significance of the host signals in sRNA expression. A new term called the complex regulatory network therefore emerged because a virulence factor could be regulated by other regulators. Since then, in vivo sRNA-expression patterns have been measured with different approaches, such as RT-qPCR and, more recently, RNA-Seq. Because the concept of sRNAs is still recent, there is limited data on the expressions of *S. aureus* sRNAs in vivo. Overall, RNAIII is the most studied sRNA.

A key step in monitoring sRNA expression is to link the expression and severity of the infection. Initially, the analysis of the expressions of five sRNAs in humans at different pathophysiological stages (skin or cystic fibrosis infections and nasal colonization) did not show any correlation between the sRNA levels and these patterns [138]. However, the authors observed that the sRNA expression levels were rather similar and uniform in nasal colonization, but not during infection. This suggests that stress is moderate during nasal colonization, and that *S. aureus* are able to adapt to this main ecological niche. In 2016, it was proposed to use the RNAIII/SprD ratio as a marker of severity [139]. This ratio enabled researchers to differentiate the strains responsible for nasal colonization and sepsis. Of note, these premise results were encouraging, but the data were obtained after in vitro cultures, and so it appears complicated to extend in vitro data to in vivo data. We recently proposed to use only SprD as a marker of severity because it demonstrated a high correlation between its expression and mortality in the *G*. *mellonella* model [88].

We already knew that the expressions of virulence factors were different between in vitro conditions and in vivo models. Numerous studies attest to the different behavior of *S. aureus* between in vitro and in vivo environments. Burian et al. demonstrated an increase in the expression of adhesins and immune-evasion factors during nasal colonization in humans and cotton rats [140,141]. The study of the *S. aureus* transcriptome in human skin abscesses showed that 262 genes had increased expressions, and that 190 were decreased [142]. Recently, similar results reinforced this finding, with the RNA-Seq data showing differential patterns between in vitro planktonic states and humans or animals, whether in the colonized or infected stages [81,143,144,145]. These results are not surprising because as soon as bacteria are in contact with a host, they must counteract different physicochemical conditions. These include aggression by the immune system and competition with the microbiome, which require the adaptation and modulation of gene-expression patterns. These outcomes imply the major role of regulators, including sRNAs. Using RNA-Seq, few studies have attempted to follow the RNome under conditions that mimick human *S. aureus* infections, including ex vivo and in vivo data (Table 3).

Each study shows significant variations when compared with the in vitro control conditions. In the first study, which monitored the in vivo expressions of several *S. aureus* sRNAs, the authors observed high expression levels of housekeeping sRNAs in a murine osteomyelitis-infection model [143]. Curiously, RNAIII and RsaA were decreased in vivo, both in the acute and chronic infection stages, while RsaC was increased. After mapping the sRNA transcriptome of the USA300 strain, Carroll et al. observed different expression levels of around 80 sRNAs in human serum compared with an in vitro culture medium [24]. Sorensen et al. demonstrated that 122 sRNAs were upregulated in vivo in a cystic fibrosis lung, and 60 in a murine vagina model, with the common feature of 29 sRNAs upregulated in both conditions [3]. Similarly, 34 sRNAs presented statistical in vivo differential expressions, either up- or downregulated [108]. Taken together, these findings reinforce the fact that sRNAs play a key role in vivo, although their precise function has not yet been determined. The main outcomes were as follows: (i) in vitro culture does not reproduce in vivo adaptation, and particularly for sRNA expression; (ii) a large pool of in vivo highly expressed sRNAs, whose functions are still unknown, deserve particular focus; (iii) some inconsistencies between the virulence implication and the in vivo-expression-pattern profile are highlighted.

Besides an overview of multiple sRNAs expressed in vivo, RNA-Seq and RT-qPCR allowed for an estimation of the sRNA expression levels of well-characterized in vitro sRNAs (Figure 4).

Among these data, two types of discrepancies can be observed: (i) inconsistences between studies, and (ii) inconsistences between the role in virulence regulation and the expression level. In some studies, sRNAs such as SprD or SprC appear more expressed in vivo than in vitro [24,88], while, in others, it is the opposite [3]. Many factors, such as the strain’s genetic background or the animal model tested, are different from one study to another. For instance, the Newman strain differs with a mutation in SaeS, a global regulator implicated in virulence regulation, which is the source of a particular exoprotein profile, in comparison with the HG003 strain [146]. The *S. aureus* N315 strain is *agr*-deficient and has frequently been used to identify sRNAs [12,17,19,20,22]. This fact endorses a different virulence-factor pattern, which probably also affects sRNAs. Therefore, it makes sense that the sRNA expression might be different between two strains whose genetic backgrounds are distant. The chosen infection model can be crucial to prove the relevance of a virulence factor [147]. This is true for the Panton–Valentin leucocidin (PVL), which may be responsible for severe necrotizing infections [130]. The study of the PVL function has been controversial for a long time, as some authors did not find the expected effect in infection models such as rats and mice, as opposed to humans and rabbits [148,149]. The difference was that the PVL receptor is sensitive to this toxin in humans and rabbits, but not in the other two. Cheung et al. expressed their opinion about the choice between rabbits and mice to study and analyze the impact of the *agr* system, explaining that the rabbit model is a better alternative based on the facts previously mentioned [83]. They also demonstrated that the *agr* system (and so RNAIII) affected the mortality in a mouse sepsis model, whereas it did not in a rabbit sepsis model, which could mean that RNAIII was either not expressed or not effective.

Overall, most studies that measured the RNAIII expression level are in line with an in vivo downregulated expression [3,88,136,137,138,143]. Others found similar results with components of the *agr* system [81,140,141,150,151]. This pattern is partially understood during nasal colonization, where the *agr* system is not expressed [140,141]. *S. aureus* would modulate the expression of its virulence-factor arsenal in favor of the expression of adhesins and immune-escape factors, which explains, in part, the weak level of *agr* expression. Similar results were observed during skin colonization [150]. The authors speculated that other staphylococci could inhibit the *agr* system, which is a likely hypothesis because a coculture of *S. aureus* and *S. epidermidis* altered the expression of at least one *S. aureus* sRNA [152]. Another possibility would be the presence of hemoglobin in the nose, which would inhibit the *agr* system, thereby promoting nasal colonization [153]. As mentioned before, the pathophysiology of *S. aureus* infections, and the concomitant role of RNAIII, are still quite complex to decipher and could explain the so-called “RNAIII paradox” that some research teams mentioned [8]. In cases of *S. aureus* persisters or SCV phenotypes, growth and metabolism are arrested [154,155]. This could explain the inactivity of the *agr* system and RNAIII because quorum sensing would not activate it. In a recent study comparing the transcriptome of wild-type and SCV *S. aureus* strains, the results were surprising because they showed that the *agr* locus and associated virulence genes were upregulated in the SCV phenotype [156]. These contradictory results highlighted obvious missing links in the total understanding of complex regulatory networks. Another paradox relies on the in vivo biological function of SprD. While it was demonstrated that the sRNA represses the expression of Sbi, an immune-evasion factor, which suggests that the deletion of SprD would increase the pathogenicity, systemic infection in mice and in a *G. mellonella* model showed the opposite [33,88]. This indicates that the variations in the SprD expression observed in vivo by others are sufficient to remodel its interactome in favor of a more offensive mode. This example highlights that the sole study of sRNA expressions in vivo might be inadequate to clearly understand their contribution.

A weak and downregulated in vivo expression was also observed for RsaA [3,88,138]. This is not surprising because RsaA promotes persistence and biofilm formation [157], and the expression levels were obtained from acute infections and not from biofilm infections. In contrast, in a chronic murine osteomyelitis-infection model, the RsaA was still downregulated, although the authors specified that *S. aureus* exhibited a persistence mode while not proving biofilm formation [143].

Broadly, the expressions of RsaC, RsaE, and RsaG were upregulated in vivo, whatever the kind of infection. These results strengthened their implication in *S. aureus* metabolism and supported the concept that metabolism and virulence may be connected. Therefore, the global function that scrutinizes RsaC, RsaE, and RsaG deserves special consideration.

The majority of sRNAs are not expressed constitutively, but rather respond to the specific stresses that prevail in a microenvironment [158]. It is described that the *agr* system was underexpressed in many human niches in response to variations in environmental signals [159]. Indeed, different environmental conditions alter the *S. aureus* behavior and compete with the *agr* system [160]. These considerations could lead to an irregular sRNA expression between colonization and infection, either acute or chronic. This was partly proven with the differential sRNA expressions in cystic fibrosis lung patients and in a murine vaginal colonization [3]. Concluding remarks about sRNA expressions, and extrapolations about their role, must be discussed carefully, and must consider the points outlined above. Moreover, monitoring the sRNA expression in selected microenvironments would be an additional approach to further examine the sRNA implication in staphylococcal regulation, which relies on the coordinated expression and activity of the whole set of regulators that constitute the regulome.

## 6. sRNA Interconnections in the Complex Regulatory Network

*S. aureus* sRNAs are mainly regulators that act at the post-transcriptional level. However, the control of gene expression is also dependent on protein factors, such as TFs, TCSs, and sigma factors.

These have been extensively studied for their role in the regulation of virulence or metabolic reprogramming [114,160,161]. While the study of sRNAs as key players of bacterial adaptation is more recent, the data discussed so far reveal that sRNAs, TCSs, TFs, and sigma factors are all embedded in global regulation, with either synergistic or antagonist roles. In the next subsections, the link between these factors will be discussed, with a focus on sRNAs as targets or regulators of protein regulators.

### 6.1. The Role of Sigma Factors in Regulatory Network and sRNA Expression

Sigma factors are subunits of RNA polymerase (RNApol) that recognize specific promoter motifs to initiate transcription. They allow the fast and effective reprogramming of the transcriptome as a function of the growth phase and global environment changes [162]. Sequence homologies, using *Bacillus subtilis* as a template, showed the presence of four sigma (Sig) factors in *S. aureus*: Sigma A, Sigma B, Sigma H, and Sigma S [163,164,165].

SigA is considered as the housekeeping (or primary) sigma factor that enables transcription under optimal growth conditions [163]. The other sigma factors are alternative factors because they normally respond to different environmental cues. The alternative stress factor Sigma B (SigB) requires the expression of four genes (*rsbU*, *rsbV*, *rsbW,* and *sigB*), organized in an operon and constitutively expressed under optimal growth conditions via SigA. During stress, SigB is released, which allows the recruitment of RNApol and binding onto DNA to promote the gene expression of its regulon [166]. Besides its global contribution in virulence [167,168], SigB is involved in heat resistance [169], acid resistance [170], and antibiotic tolerance [171]. It also confers protection against oxidative stress (e.g., through the production of Staphyloxanthin), which leads to resistance against reactive oxygen species (ROS) produced by the host’s immune system, which can be responsible for the appearance of chronic infections [171]. Additionally, it is known to regulate biofilm formation [172] and the bacterial intracellular persistence of the bacteria within host cells through the formation of SCVs [173,174].

Overall, around 250 SigB-dependent targets were identified using both 2D-DIGE or microarray analyses, and similar consensuses were proposed [175,176]. While the initial reports focused on the gene-coding content, some studies aimed at expanding the SigB regulon to sRNAs, along with the burst in the *S. aureus srna* gene discoveries. It all started in 2009 with the identification of Rsa sRNAs, with three of them being reported as SigB-dependent (RsaA, RsaD, and RsaF) [18]. These findings showed that the SigB regulon encompasses both protein-encoding genes and sRNA-encoding genes, as described for *Listeria monocytogenes* [177]. Using a bioinformatical approach, others added three novel sRNAs (SbrA, SbrB, and SbrC) to the SigB regulon [20]. They selected *srna* genes of interest based on the presence of a SigB consensus, an intrinsic downstream transcriptional terminator, and the absence of an ORF, all within IGRs. The SigB-dependent pattern was further validated by Northern blot, although direct interaction was not proven.

A large transcriptomic study accompanied by computational analyses provided an extensive repository of *S. aureus* genes (coding and noncoding) expressed in the strain HG001 [25]. Among all the genes differentially expressed under conditions that mimic the laboratory or infection, 145 putative SigB-dependent genes (86 coding genes and 59 *srna* genes) were identified with a SigB consensus upstream of their transcription start sites. Conversely, 1269 genes were predicted as SigA-dependent, including 464 putative *srna* genes identified by the authors, with some of them being described in the SRD database [14], or in the bona fide sRNA list proposed by Liu and coauthors [28]. An analysis and comparison with sRNAs detected by RNA-Seq in the SRD database suggest that only three *srna* genes contain a SigB binding site, including the bona fide SigB-dependent sRNA RsaA [18,28]. On the one hand, the SigB-dependent sRNA RsaD, for which the direct or indirect nature of the regulation by SigB remains to be experimentally determined, was not retrieved [51]. On the other hand, up to 42 sRNAs can be attributed to the direct SigA regulon, including the bona fide RsaD, RsaE, RsaH, RsaOG, and SprD sRNAs [28], or the antivirulent sRNA SprC [34]. Additionally, members of the Teg and Sau families, for which regulatory functions remain to be elucidated [12,23], mostly harbor SigA-dependent profiles. Hence, because the expression of sRNAs appears to be mainly dependent on the binding of SigA onto their promoters, the modulation of their expressions in response to stress is likely directly controlled by TCSs and TFs, rather than by the SigB general stress factor alone.

### 6.2. Two-Component Systems and Their Control over sRNA Expression

TCSs are composed of a membrane-bound sensor histidine kinase that senses external environmental stimuli, and of a response regulator, which regulates the target gene expression. They are critical for quorum sensing and adaptation to a wide range of environmental stresses, including antibiotic challenges.

So far, 17 TCSs are described in *S. aureus*, with 16 TCSs being part of the core genome, and the last one being encoded in the SCC*mec* element [178]. Among them, only WalKR is essential for bacterial growth [179]. Because *S. aureus* TCSs are already extensively reviewed elsewhere [178,180,181], only TCSs for which a clear link with sRNA regulation is reported will be described (Table 4).

Five TCSs are reported to activate or repress sRNA transcription. SaeRS, which is involved in the regulation of virulence factors, is also described as an activator of RNAIII expression [182]. On the one hand, SrrAB, which is also known to regulate virulence-factor expression under anaerobic conditions, acts as a repressor of RNAIII [11]. On the other hand, it induces RsaE and RsaD expressions, which are sRNAs for which a link with virulence has not yet been reported [39]. This suggests that SrrAB may encompass other functions in *S. aureus*. AgrAC, which is involved in the regulation of quorum sensing and the activation of PSMs [183], is also responsible for the repression of ArtR sRNA [68]. ArlRS, a regulator of autolysis, acts negatively on RNAIII expression [184], which indicates that several TCSs tightly monitor the expression of RNAIII. Finally, HptRS is involved in the adaptative response to fulfill a particular carbon source, and it is an activator of RsaG expression [39].

### 6.3. Transcription Factors and Their sRNA Regulons

TFs are the largest class of regulators. They play a critical role in favoring and/or impeding the activity of the RNA polymerase through the recognition of DNA sequences on the promoter region [196]. In *S. aureus*, Ibarra and colleagues estimated the presence of 114 TFs [197]. However, as for sRNAs, only a small number have been thoroughly characterized. Those that regulate sRNA expression represent less than 10% of the TFs predicted (Table 4). Among them, the staphylococcal accessory regulator SarA is well studied and was shown to directly regulate 11 sRNAs, including RNAIII, Srn_3610_SprC, RsaD, and some sRNAs that are part of the toxin–antitoxin system, such as SprG2 or SprA2_AS_ [86,186,187,188]. In their recent study, Oriol and colleagues combined the use of ChIP-seq and RNA-Seq to conclude that the SarA sRNA regulon may encompass 51 sRNAs [188]. Although some of them may not be direct targets, it is possible that the number of direct sRNA targets is underestimated.

The CodY pleiotropic repressor is also known to directly control the expressions of several sRNAs. Interestingly, it shares with SarA the regulation of RNAIII and RsaD sRNAs [51,198], which indicates that the SarA regulon and CodY regulon overlap. This overlapping does not occur only at the sRNA level, as CodY and Rot regulate the TCS SaeRS, which highlights the complexity of intertwined regulatory pathways [199].

Other TFs that monitor the expressions of sRNAs are CcpA, SarT, MgrA, SarU, SarV, and MntR. Apart from MntR, all of them exert regulatory control over RNAIII [117,191,192,193,194]. It is noteworthy that seven of these eleven TFs regulate RNAIII, which indicates that this sRNA belongs to a critical node in the *S. aureus* regulatory network. Finally, MntR was shown to repress RsaC expression, as this sRNA is released from the 3′UTR of the *mntABC* operon [40].

### 6.4. sRNAs as Regulators of TCSs, TFs, and Other sRNAs

As discussed earlier, sRNAs are implicated in many cellular processes, such as metabolism, antibiotic resistance, and virulence, through the regulation of a large variety of targets. Besides their post-transcriptional role in the mRNA-encoding functions listed above, they also interact with mRNAs that encode TFs, TCSs, or other sRNAs. At least 10 sRNAs are reported to regulate TCSs, TFs, or sRNA expression. As for TFs, overlaps occur between RsaA and RNAIII: both regulate *mgrA*, with RsaA being a repressor and RNAIII an activator [67,157]. Additionally, RNAIII is also known to repress the expression of Rot [62]. Finally, ArtR represses SarT [68], whereas RsaC targets SarA, although the biological significance of this interaction was not investigated [36].

Some sRNAs were shown to regulate the expressions of TCSs. This is the case for Teg49 and SSR42, which both target SaeRS [121,127]; RsaE, which acts on SrrAB [200]; and SprX, which targets the essential TCS WalKR [69]. Moreover, RsaG was shown to interact with *sarA* and *tcaR* [42]. Whereas in vitro experiments further validated these interactions, the authors did not observe changes in the SarA or TcaR protein quantity upon the deletion of RsaG, which indicates that the interconnections are probably more complicated than initially anticipated.

Some studies indicated interactions between sRNAs that suggest that some sRNAs act as sponges. A newly characterized sRNA, SprY, was shown to titrate RNAIII [41]. RsaI was also shown to interact with RsaG, although the precise mechanism of action has not yet been uncovered [39]. It seems to be the center of a regulatory loop between sRNAs, which is able to also interact with RsaD and RsaE. However, the regulatory role of these interactions needs to be elucidated.

Altogether, this indicates that the nodes between TCSs, TFs, sRNAs, and their classical targets are extremely complex, and that sRNAs are important actors. Few sRNAs emerged as nodes of particular interest because many intertwined connections can be raised between the classes of regulators and biological functions already identified (Figure 5). This is the case for RNAIII, whose expression is tightly regulated (activated or repressed) not only by multiple TCSs and TFs, but also by the sponge RNA SprY. Therefore, these regulators play an indirect role in RNAIII direct targets, among which, two encode TFs (*rot* and *mgrA*) involved in *S. aureus* virulence. The *mgrA* mRNA is also a target of the antivirulent sRNA RsaA, whose regulatory role is opposite to that of RNAIII. Additionally, the SarA and CodY TFs and the SrrAB TCS all regulate the expressions of RNAIII and RsaD, which suggests that these two sRNAs need to be under certain coregulated conditions in order to adapt to the *S. aureus* physiology. On top of this, studies on RsaD and RsaA expressions revealed that they belong to the SigB regulon, which increases the complexity of the regulatory network. It is probable that the knowledge gained so far only represents the tip of the iceberg, and especially considering that RsaI has the potential to come under the spotlight of regulatory loops.

## 7. Concluding Remarks

Since the 1990s, the discoveries and characterizations of staphylococcal sRNAs have brought another facet to the *S. aureus* conception of regulation and adaptation. At least more than one hundred are described, with functions and modes of action that are constantly evolving, as evidenced by the dual-function sRNAs, RNA sponge activity, and sRNA cargo of extracellular vesicles. One of the issues lies in the difference between the high number of sRNAs discovered and the sparse number characterized. To date, about fifteen sRNAs are known to regulate metabolism and/or virulence. Even though their physiological relevance and connection to other major regulators have been demonstrated, the rise of the sRNAs has subsided somewhat. The fact that no crucial or inconsistent effects related to the virulence were demonstrated may have tempered researchers, but they can be explained nowadays. An sRNA-deficient strain is not necessarily more or less virulent because all the regulators are embedded in the complex regulatory network, and another regulator can replace it, which thus creates a switch in this network. Strain-dependent, animal-dependent, or microenvironment-dependent effects also complicate a thorough and in-depth understanding of the precise roles of sRNAs. Priorities must tend towards in situ experiments within the host, given the abovementioned specifics. The most striking illustration concerns the *agr* system and its effector, RNAIII, which are considered to be the major regulators of virulence, but with low in vivo expression profiles. Deciphering the sRNA involvement should no longer be performed independently, but by using a global approach to connect all the implicated actors. The expansion and utilization of high-throughput technologies directly within the host are a source of renewed interest and allow for a sharper cartography of the complex regulatory networks. In vivo RNA-Seq may uncover a part of the hidden face, but it will not provide sRNA direct targets. Recently, MAPS and CLASH have considerably modified the situation, although the experiments were realized in vitro. In *Salmonella*, the MAPS was successfully used in an ex vivo macrophage model to gain insight into the virulence network, which suggests that it could also be used for *S. aureus* [201]. Besides colonization, it is obvious that *S. aureus* infections denote multiple different diseases. Consequently, different regulations and adaptations between biofilm communities, abscesses, and intracellular lifestyles must also be taken into consideration. Finally, the majority of sRNA experiments emphasized an *S. aureus* adaptation to its host, but the opposite must also be integrated, which means that a specific immune-system trait could enable better insight into the complex regulation network. Beyond the approach to the physiopathology, finality would be to glimpse novel perspectives for regulatory-mediated therapeutics, as recently proposed by others [202].

## Figures and Tables

**Figure 1 ijms-23-07346-f001:**
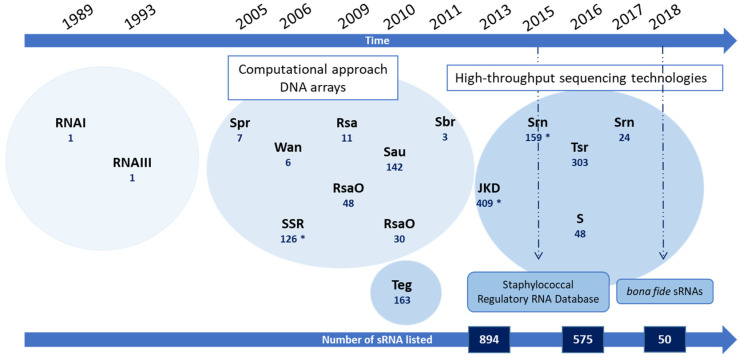
Chronology and classification of the major classes of *Staphylococcus aureus* sRNAs discovered from the 1990s to nowadays. * indicates that there is no further validation after the identification of a new sRNA group.

**Figure 2 ijms-23-07346-f002:**
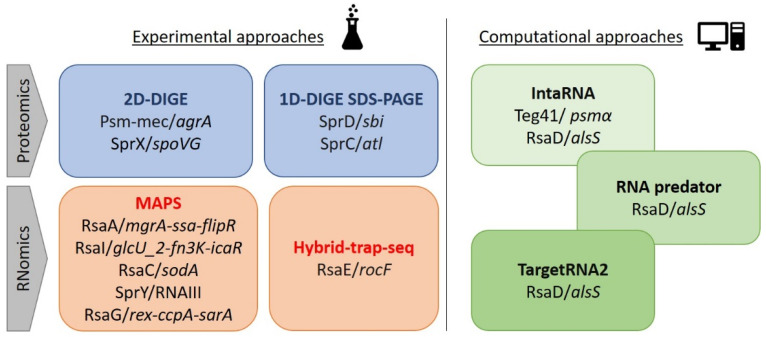
Main experimental and computational approaches used to identify RNA targets of *Staphylococcus aureus* sRNAs. On the left side are presented experimental approaches based on the identification of RNAs using proteomics or RNomics. On the right side are listed computational approaches. For both methods, examples of sRNAs and their targets are indicated.

**Figure 3 ijms-23-07346-f003:**
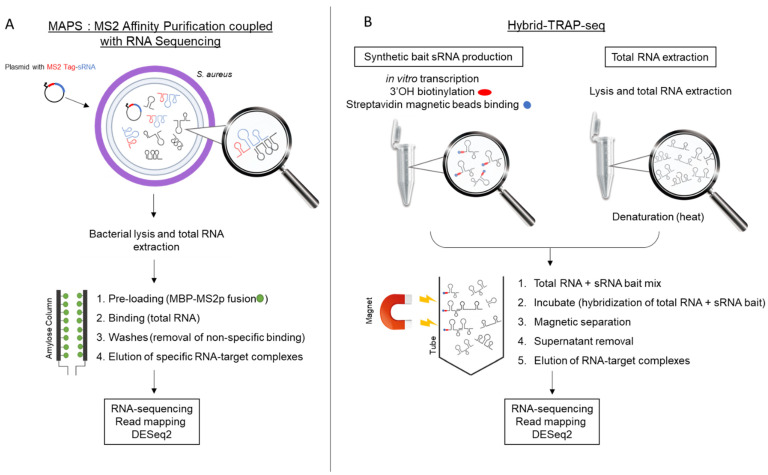
Experimental high-throughput techniques to identify an sRNA interactome in *S. aureus*: (**A**) MAPS (MS2-affinity purification coupled with RNA sequencing) protocol. Green dots: MBP (maltose-binding protein) in fusion with MS2p (MS2 protein); (**B**) Hybrid-trap-seq protocol. Red ovals: biotinylation in the 3′OH end of bait sRNA. Blue dots: streptavidin magnetic beads bound to the biotinylate 3′OH ends of bait sRNA.

**Figure 4 ijms-23-07346-f004:**
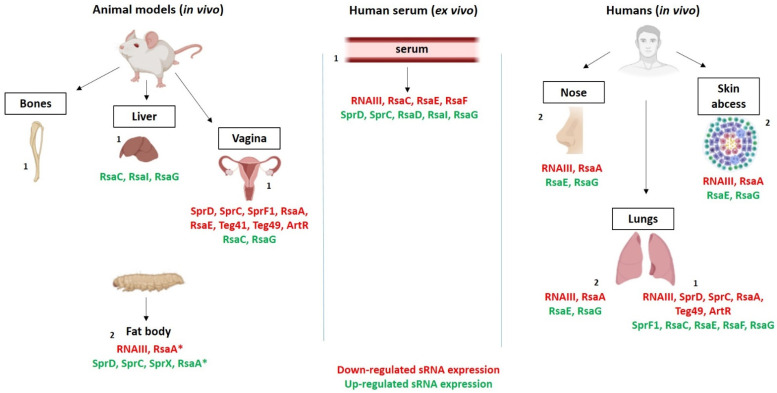
In vivo or ex vivo expressions of *Staphylococcus aureus* sRNAs contributing to the regulation of metabolism and/or virulence. In vivo or ex vivo expressions are compared to in vitro calibrators. Numbers 1 and 2 correspond to sRNA expressions obtained by RNA-Seq and RT-qPCR, respectively. * indicates in vivo down- or upregulation, according to the exponential or stationary growth phases [3,24,88,108,138,143].

**Figure 5 ijms-23-07346-f005:**
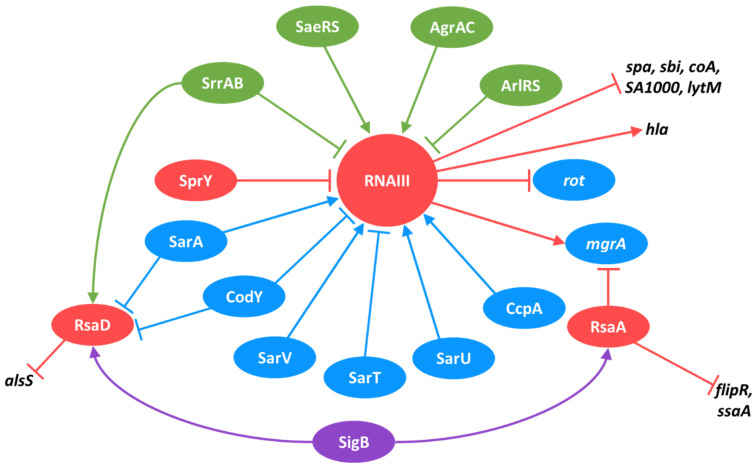
Specific example illustrating the complexity of *Staphylococcus aureus* regulatory network. RNAIII constitutes a central node along with RsaA and RsaD, more recently defined. sRNAs, two-component systems, transcriptions factors, and sigma regulators are represented in red, green, blue, and purple, respectively. mRNA virulence factors are represented in black. The plain lines indicate direct regulation, with arrows corresponding to activation, and bars to repression.

**Table 2 ijms-23-07346-t002:** Important features of sRNA characterized in *Staphylococcus aureus*. ORF: open reading frame; N/A: not applicable.

Name	Consensual Name	Length (nt)	ORF	Direct mRNA Targets	Mechanisms of Action	Function
**RNAIII**	Srn_3910	514	Yes (Hld)	*spa*, *sbi*, *coa*, *sa1000*, *lytM*, *rot* *mgrA*, *hla*	Translation inhibition (*lytM*, *sbi*), translation inhibition and mRNA cleavage (*rot*, *spa*, *coa*, *SA1000*), translation activation (*hla*), mRNA stabilization (*mgrA*)	Provirulent
**Psm-mec**	N/A	143–157	Yes (Psm-mec)	*agrA*	Translation inhibition	Antivirulent
**SprC**	Srn_3610	154	No	*atl*	Translation inhibition	Antivirulent/Provirulent
**SprD**	Srn_3800	145	No	*sbi*	Translation inhibition	Provirulent
**SprF1**	Srn_3830	138	No	*sprG1*, ribosomes	Translation attenuation	Persistence
**SprX**	Srn_3820	150	No	*spoVG*, *walR*, *ecb*, *clfB*, *hld*	Translation inhibition (*spoVG*, *ecb*)	Provirulent
**SprY**	Srn_9630	128	No	*RNAIII*	Seric blocking of mRNA binding sites	Antivirulent
**RsaA**	Srn_1510	143	No	*mgrA*, *flip-r*, *ssaA*	Translation inhibition	Antivirulent
**RsaC**	Srn_1590	Strain-dependent	No	*sodA*, *sarA*	Translation inhibition (*sodA*)	Provirulent, metabolism
**RsaD**	Srn_1640	177	No	*alsS*	Translation inhibition	Metabolism
**RsaE**	Srn_2130	459	No	*rocF*	Translation inhibition	Metabolism
**RsaF**	N/A	105	No	*hysA*, *splD*	Unknown	Undefined
**RsaG**	Srn_0510	194	No	*rex*,		Metabolism
**RsaI**	Srn_4390	111	No	*glcU_2*, *fn3K*, *icaR*, *rsaG*	Translation inhibition (*glcU_2*, *fn3K*, *IcaR*)	Metabolism
**ArtR**	Srn_4050	346	No	*sarT*	Translation inhibition and mRNA degradation	Undefined
**SSR42**	Srn_4470	1232	Yes (unknown peptide)	*sae*	Unknown (mRNA stabilization?)	Provirulent
**Teg49**	Srn_1550	196	Yes (unknown peptide)	*sarA* *spn*	mRNA stabilization	Undefined
**Teg41**	Srn_1080	205	Yes (unknown peptide)	*psmα*	Unknown (mRNA stabilization or translation initiation?)	Provirulent

**Table 3 ijms-23-07346-t003:** sRNA expressions in conditions mimicking human *Staphylococcus aureus* infections. Expressions were obtained by RNA-Seq, including in vivo and ex vivo conditions.

	Murine Osteomyelitis	Human Serum	Human Cystic Fribrosis	Murine Vaginal Colonization	Murine Liver
**Conditions**	In vivo	Ex vivo	In vivo	In vivo	In vivo
**sRNA expression**	>15 differentially expressed sRNAs	42 upregulated 41 downregulated	122 upregulated	60 upregulated	17 upregulated 17 downregulated
**Comparator**	BHI medium	TSB medium	Chemically defined medium, synthetic fibrosis media	Laboratory media	TSB medium
**Kinetics**	Yes Acute infection (7 days), chronic infection (28 days)	No	No	Yes 5 h, 24 h, 72 h	Yes 6 h, 24 h, 48 h
**References**	[143]	[24]	[3]	[3]	[108]

**Table 4 ijms-23-07346-t004:** Transcriptional regulation of sRNA expression by two-component systems and transcription factors. (−) indicates negative regulation and (+) indicates positive regulation.

Regulator	Name	Functions	sRNA Targets	References
**TCSs**	SaeRS	Regulation of virulence factors	RNAIII (+)	[160,182]
	SrrAB	Oxydative stress	RsaE (+), RsaD (+), RNAIII (−)	[11,160]
	AgrAC	Regulation of virulence factors, activation of quorum sensing	RNAIII (+), ArtR (−)	[68,183]
	ArlRS	Autolysis regulation	RNAIII (−)	[184]
	HptRS	Hexose phosphate transport	RsaG (+)	[39,185]
**TFs**	SarA	Global regulator of virulence determinant	Many sRNAs, including RsaD (−), sprG2 (−), Spr2AS (−), SprC (−), Srn_9340 (−), RNAIII (+)	[86,132,186,187,188]
	CodY	Adaptive response to starvation, regulation of virulence factors	Many sRNAs, including RsaD (−), RNAIII (−)	[51,189,190]
	CcpA	Adaptive response to carbon source, modulation of virulence factors	RsaI (−), RNAIII (+)	[39,117]
	SarT	Repressor of alpha hemolysin synthesis	RNAIII (−)	[132,191]
	MgrA	Global regulator of virulence factors	RNAIII (+)	[192]
	SarU	Positive regulator of agr	RNAIII (+)	[193]
	SarV	Autolysis regulator	RNAIII (+)	[194]
	MntR	Control of manganese uptake	RsaC (−)	[40,195]

TCSs: two-component systems; TFs: transcription factors.

## Data Availability

Not applicable.

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
