# Peer review of "Thirty Years of sRNA-Mediated Regulation in Staphylococcus aureus: From Initial Discoveries to In Vivo Biological Implications"

_ijms, 2022, doi:10.3390/ijms23137346_

Round 1

Reviewer 1 Report

A very well-written review by Menard et al. describes an interesting field and the data is very well presented. However, there are some places where the English language would need an improvement (lines 88, 144).

It would be helpful if the authors created graphics instead of long tables. That way, the information would be easier to follow.

Author Response

We thank the reviewer for his (her) expertise and for stating the review is well-written and interesting.

We proceeded to a thorough reading of the manuscript to improve English language, especially the lines cited by the reviewer. These appear on line 105 and line 189 of the new version of the manuscript. We also did some modifications elsewhere.

We also thank the reviewer for stating that graphics (instead of long tables) would be ideal for a better understanding . Therefore, we changed two tables in figures (Figure 2 and 4 of the current revised version).

Reviewer 2 Report

In this review article the authors present a historical perspective of the progress of sRNA research in the pathogen Staphylococcus aureus. The authors undeniably bring a welcome rationalization and gather here a huge amount of information. In this sense, a review on this subject is a valuable work that deserves publication, especially since no equivalent review is available to my knowledge. 

That said, the reading of this review remains, for the non-specialist, arduous. This is largely because the introduction is not detailed enough to allow one to enter the subject and does not announce the structure of the body of the manuscript. As it stands, the specialist reader will certainly be able to pick out useful elements, but the novice reader will only see a juxtaposition of historical summaries, not linked to each other. Similarly, the conclusion is much too brief for my taste, whereas it should put into perspective the considerable mass of information presented here.

This review article will have the impact it deserves only if the introduction, the conclusion, and the links between the different parts are reconstructed for a fluent reading by non-specialists.

Author Response

We thank the reviewer for the kind comments.

Based on suggestions, we extensively edited the introduction to provide more information for non-specialists. We also clarifly the structure of our review by providing the key elements of our plan at the end of the introduction. We also attempted to provide better links between the differents part so that the reader should not see anymore a juxtaposition of historical summaries but a well-balanced story of current knowledge. Finally, we substantially edited the conclusion to provide more information.

Reviewer 3 Report

This review is devoted to the regulatory RNAs of such a socially significant pathogen as Staphillococcus aureus. The work turned out to be multifaceted and contains both a retrospective analysis with a description of experimental and computer approaches and methods used to identify and characterize various sRNAs, as well as a fairly detailed characterization and description of the role of the main representatives of regulatory RNAs. References are made to in vitro, in vivo, and in silico studies, and finally, a discussion and analysis of the complex regulatory networks that comprise sRNA is provided. The review seems to be very elaborate, read "in one breath" and deserves to be published. Minor technical errors can be corrected in proofreading mode.

Author Response

We thank the reviewer for this kind review of our manuscript. We did a thorough reading of the manuscript to remove the technical errors of the previous version.